# Impact of systematic medication review in emergency department on patients' post-discharge outcomes—A randomized controlled clinical trial

Lisbeth Damlien Nymoen[1,2]*, Trude Eline Flatebø[1], Tron Anders Moger[3], Erik Øie[4], Espen Molden[2,5], Kirsten Kilvik Viktil[1,2]

1 Diakonhjemmet Hospital Pharmacy AS, Oslo, Norway, 2 Department of Pharmacy, University of Oslo, Oslo, Norway, 3 Department of Health Management and Health Economics, University of Oslo, Oslo, Norway, 4 Department of Internal Medicine, Diakonhjemmet Hospital, Oslo, Norway, 5 Department of Psychopharmacology, Diakonhjemmet Hospital, Oslo, Norway

* lisbetd@student.matnat.uio.no

**Data Availability Statement:** The data that support the findings of this study are not publicly available, because of the content of sensitive patient information and indirect patient identifiers,

## Abstract

### Introduction

The main objective of this study was to investigate whether systematic medication review conducted by clinical pharmacists can impact clinical outcomes and post-discharge outcomes for patients admitted to the emergency department.

### Method

This parallel group, non-blinded, randomized controlled trial was conducted in the emergency department, Diakonhjemmet Hospital, Oslo, Norway. The study was registered in ClinicalTrials.gov, Identifier: NCT03123640 in April 2017. From April 2017 to May 2018, patients ≥18 years were included and randomized (1:1) to intervention- or control group. The control group received standard care from emergency department physicians and nurses. In addition to standard care, the intervention group received systematic medication review including medication reconciliation conducted by pharmacists, during the emergency department stay. The primary outcome was proportion of patients with an unplanned contact with hospital within 12 months from inclusion stay discharge.

### Results

In total, 807 patients were included and randomized, 1:1, to intervention or control group. After excluding 8 patients dying during hospital stay and 10 patients lacking Norwegian personal identification number, the primary analysis comprised 789 patients: 394 intervention group patients and 395 control group patients. Regarding the primary outcome, there was no significant difference in proportion of patients with an unplanned contact with hospital within 12 months after inclusion stay discharge between groups (51.0% of intervention group patients vs. 53.2% of control group patients, p = 0.546).

imposed by the Regional Committee for Medical and Health Research Ethics (REC) (2015/1356/ REK south-eastern A). Data may be obtained from a third party, Diakonhjemmet Hospital (contact via e-mail: postmottak@diakonsyk.no) on reasonable request, but restrictions apply to the availability of these data, which were used under license for the current study, and thus cannot be made publicly available. Deidentified participant data are available from the authors on reasonable request and with permission of Diakonhjemmet Hospital and REC (contact via e-mail: rek-sorost@medisin.uio.no).

**Funding:** The study was funded by Diakonhjemmet Hospital Pharmacy AS, Diakonhjemmet Hospital fund for research, innovation and professional development and Diakonhjemmet Foundation. The funders had no role in study design, data collection and analysis, decision to publish, or preparation of the manuscript.

**Competing interests:** The authors have declared that no competing interests exist.

## Conclusion

As currently designed, emergency department pharmacist-led medication review did not significantly influence clinical- or post-discharge outcomes. This study did, however pinpoint important practical implementations, which can be used to design tailored pharmacist-led interventions and workflow regarding drug-related issues in the emergency department setting.

## Introduction

Over several decades transitions of care, for instance admission to an emergency department (ED), have been identified as a key risk event regarding drug-related patient safety [1, 2]. More than 60% of drug histories registered by physicians in the ED does not reflect patients' actual drug use before admission [3–6]. Incomplete information about patients' drug use during transition of care can lead to medication errors and drug-related problems during and after a hospital stay [7–10]. It is reported that up to 27% of hospital prescribing errors is linked to inaccurate or incomplete ED drug histories [8]. Hence, errors occurring during the ED stay can stick with the patient during the entire hospital stay. Drug-related problems are also common in the ED, it has been reported that 85% of patients admitted to the ED has at least one drug-related problem [11]. Furthermore, drug-related problems leading to ED visits has been reported to affect up to 30% of patients admitted to the ED [12–18].

The increasing worldwide challenge with crowding in the ED [19–21], force ED physicians to prioritize their time to ensure that all admitted patients receive appropriate emergency care. In several countries obtaining drug histories is a task assigned to ED physicians, and it has been reported that ED physicians down-prioritize obtaining drugs histories when the ED is crowded [22]. In addition, a growing body of evidence suggests that ED physicians do not recognize drug-related problems and drug-related ED visits in the fast-paced workflow, which can led to misdiagnosing and treatment of the symptoms instead of the actual problem [14, 15, 23, 24].

Integrating pharmacists in ED care has been reported to increase the quality of drug histories obtained in the ED [4, 25, 26], reduce medication errors during and after hospital stay [27–29], and reduce delay of care [30]. Furthermore, systematic medication review conducted by pharmacists in the ED has been found to increase recognition of drug-related ED visits [14, 15]. Previous randomized controlled trials (RCT) investigating impact of pharmacist-led interventions on clinical outcomes have included patients from hospital wards (i.e., in-hospital patients) [31–38]. Systematic reviews meta-analyzing results from these RCTs conclude that pharmacists-led interventions conducted in hospital wards reduce medication errors, reduce subsequent ED revisits [39], decreases drug-related hospital revisits, and all-cause readmissions [26]. The interventions in some of the prior RCTs have demanded massive resources due to follow up, for instance following the patient with several medication reconciliations and medication reviews during the entire hospital stay and further contacting the patients, by phone after discharge [31–34, 38]. As hospital resources in the real-world setting is not unlimited, studies investigating more pragmatic and more implementable interventions are necessary. When initiating the presented study, no prior RCTs had investigated the impact of pharmacists-led interventions in the ED-setting on clinical outcomes. However, as length of hospital stay has decreased worldwide the last decades [40], drug-related ED visits are common, and frequent registration of incorrect drug histories in the ED is concerning, it is

important to investigate interventions conducted during the ED stay i.e., early during the hospital admission to prevent drug-related harm during and after a hospital stay [8, 9].

Prior studies did not provide certain evidence with regards to what patient groups who could benefit from an ED pharmacist-led intervention. Surgical referral reason had been reported as a risk factor for clinically relevant medication discrepancies at admission to the ED [3, 41, 42], and medical referral reason had been identified as a risk factor for drug-related hospital admissions [43]. Further, no certain evidence with regards to a relevant age cut-off was presented in prior studies. Therefore, the aim of the presented study was to perform a RCT to investigate whether implementing pharmacist-led systematic medication reviews in the ED could impact clinical and post-discharge outcomes for a general ED population.

## Methods

### Study design

This parallel group, non-blinded RCT was conducted at the ED, Diakonhjemmet Hospital, a local, urban hospital in Oslo, Norway. Patients were included consecutively in periods from 24 April 2017 until the predetermined target number of 800 patients was enrolled on 16 May 2018. Fig 1 presents the patient flow of the study.

The study was approved by the institutional review board at Diakonhjemmet Hospital and the Regional Committee for Medical and Health Research Ethics (2015/1356/ REK south-eastern A). Written informed consent was obtained from all patients before inclusion. The study was designed and reported according to the CONSORT 2010 Statement [44] (S1 Appendix CONSORT checklist). The study was registered in ClinicalTrials.gov, Identifier: NCT03123 640 in April 2017, and closed for new participants May 2018. Fig 2 gives a graphical depiction of the study design. S2 Appendix shows the original study protocol, protocol amendments, and the timeline of the study with milestones.

Data for 12 months follow-up was harvested from the Norwegian Patient Registry (18/ 12607-10), a national registry which comprise information about all hospitalizations (including both isolated ED visits and admissions with or without ED visits). Data available from the Norwegian Patient Registry is routinely reported from all Norwegian hospitals. Each patient was followed for 12 months from their inclusion stay discharge and follow-up was stopped for the last patient on 30 May 2019.

To standardize study procedures, written operation procedures were developed regarding the inclusion- and randomization procedure, the data collection process, and the intervention deliverance. All involved study pharmacists were familiar with these operational procedures before study start.

### Study setting

In Norway general practitioners (GPs) and the municipal emergency clinics have a gatekeeper function and handle less severe conditions. More severe conditions are referred from GPs or municipal emergency clinics or other healthcare personnel of the primary healthcare e.g., nursing home physicians, or paramedics, to the hospitals' EDs. The referring healthcare personnel set a tentative referral reason after assessing the patient's symptoms and conducting an initial examination (before the ED admission). Based on the tentative referral reason patients are allocated to see a physician from the Department of Internal Medicine (Medical referral reasons), or a physician from the Department of Surgery (Surgical referral reasons) at admission to the investigated ED.

In Norwegian EDs, physicians are responsible for obtaining and registering patients' drug lists at admission (drug-history taking). In some Norwegian EDs clinical pharmacists are

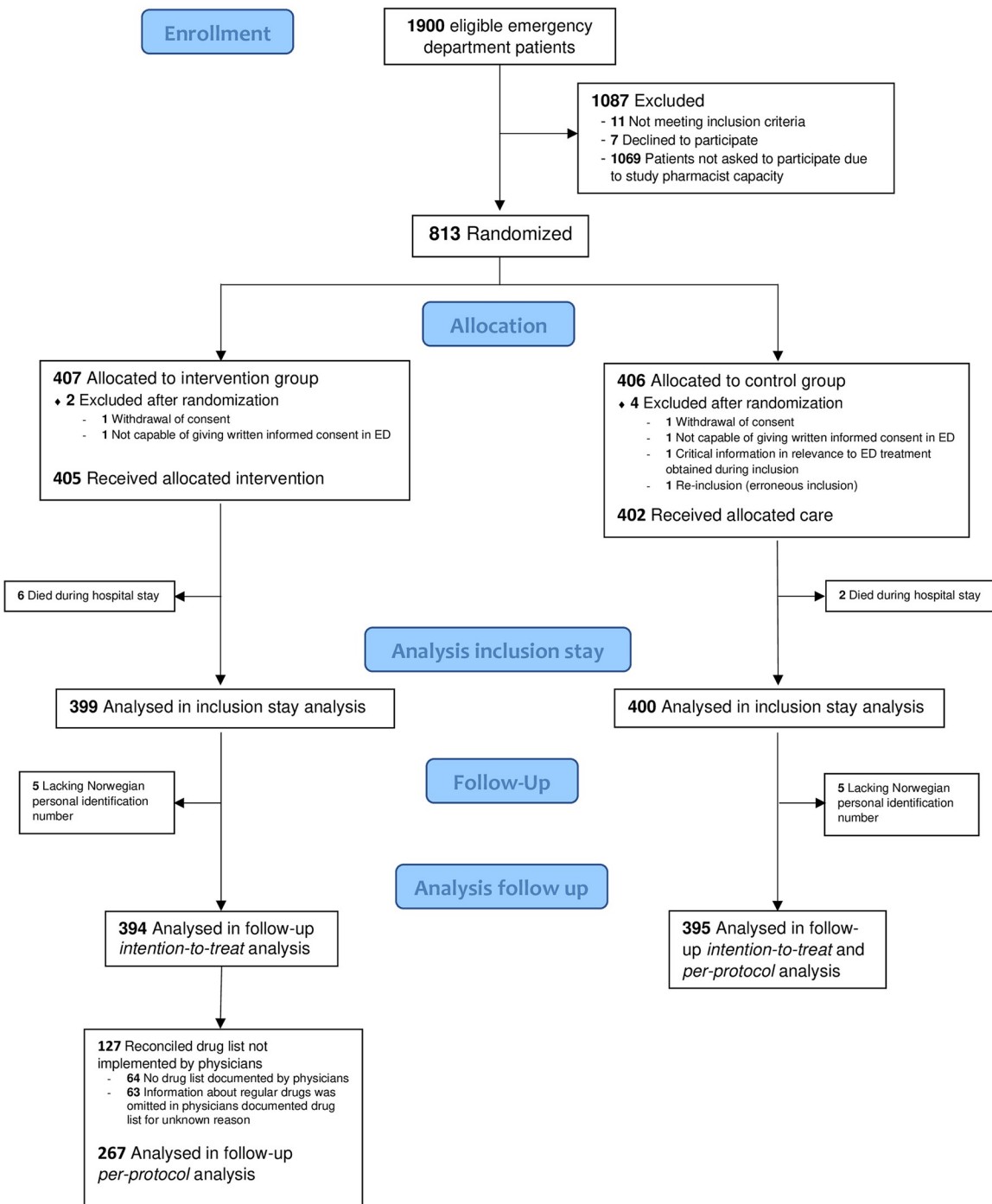

**Fig 1. Flow chart of patients eligible for inclusion in the Emergency Department (ED).**

conducting medication reconciliation and communicate their findings to ED physicians. However, this is not established as standard care, and due to limited ED pharmacist resources majority of drug histories are obtained and registered by ED physicians without a systematic medication reconciliation. In the investigated ED at Diakonhjemmet Hospital the ED is normally covered by a 0.5 full-time equivalent pharmacist position. During the data collection of

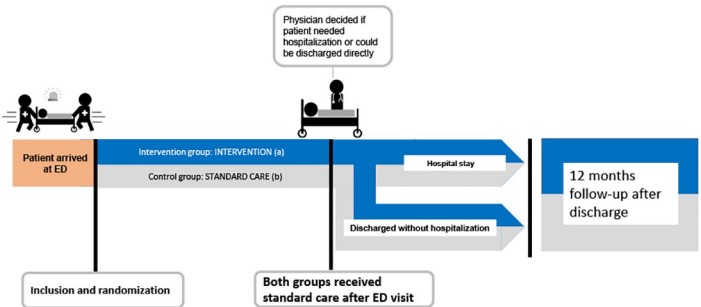

**Fig 2. Study design.** During emergency department (ED) stay the Intervention group (a) received standard care and pharmacist conducted systematic medication review (identifying drug-related problem), including medication reconciliation (obtaining patients' actual drug lists). Findings from medication reconciliation and medication review were discussed with ED physician and documented in the electronical patient record. The control group (b) received standard care (by physicians and nurses).

the presented RCT, these pharmacist resources together with additional pharmacist resources was utilized to conduct the intervention. Some Norwegian hospital wards received clinical pharmacist service, in Diakonhjemmet Hospital 15 hospital wards are covered by 4.7 full-time equivalent pharmacist positions as part of standard care.

## Participants

Annually, 13500 patients 18 years or older with both medical and gastrointestinal or orthopedic surgical symptoms are referred to the ED at Diakonhjemmet Hospital. The average length of stay in the investigated ED was 3.2 hours in 2018. All patients arriving at the investigated ED, willing to/capable of providing written, informed consent were eligible for inclusion. Unconscious patients were not included e.g., severe intoxications. Patients aged $\geq$65 with hip-fracture were admitted to a specialized ED at another location, hence these patients were not included. The patient inclusion was performed by study pharmacists working shifts according to a pre-set study schedule, either between 9:00 am and 4:00 pm or between 4:00 pm and 10:00 pm, on both weekdays and weekends.

Originally an exclusion criterion regarding terminal ill patients with short life expectancy was stated. This criterion was not feasible to meet in the fast-paced workflow of the ED. Hence, patients were included regardless of this issue. Patients readmitted during the study period were not invited for 'a second' inclusion.

## Randomization

In line with the developed inclusion and randomization operation procedure, patients were randomized to intervention- or control-group (1:1) by study pharmacists, after inclusion, i.e., after patients had given written, informed consent. Department of Biostatistics and Epidemiology at Oslo University Hospital organized the randomization sequence. This department had no contact with patients, study pharmacists or ED personnel. A random number generator program was used for randomization sequencing with a permuted block design. The study pharmacists were blinded to block size, which was randomly varied between 2–8 patients per block, average block size was 5.0 ± 2.2. Allocation information cards was packed in sequentially numbered, sealed, opaque envelopes by an independent administrator who took no further part in the study, and delivered to study pharmacists. The envelopes were assigned to each individual participant and opened in numbered sequence.

It was neither feasible to blind patients nor study pharmacists to the allocation after randomization. It was also impossible to blind the hospital staff regarding which of the patients belonged to the intervention group. Affiliates at the ED and hospital were, however, unable to distinguish between patients randomized to the control group and patients not participating in the study.

The Norwegian Patient Registry providing outcome data were blinded to group allocation.

## Intervention

The control group received standard care during the ED stay, consisting of triage-nurse consultation (often included physical measurements), consultation by physician (including physical examination, medical and medication history taking) and nurse consultation. In addition, laboratory tests and other tests (e.g., x-rays, fecal occult blood test) were taken, analyzed, and assessed by physicians during the ED stay. Regarding ethical concerns, patients from the control group were excluded after randomization if; 1) Physicians at the ED requested an assessment from a pharmacist regarding the control group patient, 2) Study pharmacist revealed obvious drug-related problems of major clinical relevance during inclusion of the control group patient and had to intervene.

The intervention group received, in addition to standard care, a systematic medication review including medication reconciliation conducted by a study pharmacist early during the ED stay. The intervention was based on the integrated medicine management (IMM)-model [45], adjusted to the fast-paced workflow of the ED-setting [3], and conducted by experienced clinical pharmacists (study pharmacists).

The medication reconciliation consisted of a standardized patient interview, including use of a checklist with specific questions about drugs often omitted, e.g., eyedrops, inhalation drugs, contraceptives, drugs not taken daily etc. If the patient received assistance with taking drugs, the supporting person/personnel was contacted to be interviewed. In addition, sources providing information on drug prescribing, e.g., electronic prescription database, drug-list of patients with multidose drug dispensing, GPs, and other hospitals, were used to verify drugs in use and the respective dosages and brand names. The medication reconciliation was preferably conducted before the ED physician consultation [3], thus the ED physicians could utilize the results from the medication reconciliation during their consultation. Due to the critically illness of some patients and occasional ED crowding the medication reconciliation was not completed before ED physicians' consultations for all included patients. The ED physician in charge of the patient was, however always alerted orally when the medication reconciliation was conducted. In addition, the complete drug list was documented in the hospital's electronic patient record by the study pharmacists.

Following the reconciliation, a systematic medication review was performed. All drugs in the reconciled list were assessed according to predefined drug-related problems categories; drug monitoring, adherence issues, adverse effects, drug-interactions, non-optimal drug therapy, unnecessary drug, based on a validated medication review-tool [46] (detailed description of the drug-related problems categories are presented in S3 Appendix). The systematic medication review was conducted by interviewing patients and assessing initial examinations performed by ED nurses as well as laboratory tests results. In addition, computer resources were utilized (e.g., interaction databases, summary of product characteristics for drugs, and medical databases), and referral letters from GPs and municipal emergency were reviewed. After identifying drug-related problems in the medication review, possible actions to manage or solve the problems were suggested by the study pharmacists and documented in the electronic patient record, as well as orally communicated to the ED physician in charge of the patient.

After discharge an interdisciplinary team consisting of two chief physicians and three experienced clinical pharmacists classified all identified drug-related problems according to clinical relevance (issues of importance for the patient treatment). The interdisciplinary team had access to the following information for all intervention group patients; demographic data, results from laboratory tests during hospital stay, tentative referral reasons, final diagnoses (documented in the discharge note by the physician discharging the patient), the patient's reconciled drug list, and identified drug-related problems. The interdisciplinary team classified the drug-related problems as clinically relevant to identify in the ED, clinically relevant to identify during the hospital stay, or not clinically relevant during the ED/hospital stay.

To investigate the efficiency of the intervention, implementation of study pharmacists' recommendations regarding the documented reconciled drug list and drug-related problems was assessed. A study pharmacist retrospectively reviewed admission notes and discharge notes written by physicians, in addition to assessing laboratory tests from the ED visit ordered according to the medication review findings. The drug lists for intervention group patients documented by physicians at admission were considered complete if there were no medication discrepancies (e.g., omissions, additions, dosage discrepancies) regarding regular drugs compared with the reconciled drug list documented by study pharmacists. Physicians' acceptance of the study pharmacists' recommendations regarding drug-related problems was determined by reviewing if changes related to the recommendations were made in the drug list during ED/hospital stay.

## Outcome measures

To investigate clinical and post-discharge impact of the intervention, objective outcome measures which were possible to measure with a high degree of precision was desirable [47], in contrast to prior ED studies investigating more subjective outcomes [27–30]. Relevant outcome measures were chosen based on outcomes in RCTs investigating pharmacist-led interventions in hospital wards [32, 37, 38]. The primary outcome measure was proportion of patients with an unplanned contact with hospital within 12 months after inclusion stay discharge (both ED revisits and hospital readmissions). This was chosen as primary outcome as only 12 months data were available for sample size calculation. S2 Appendix original study protocol and protocol amendments comprise further explanation of chosen outcomes.

Secondary follow-up outcomes were:

- Proportion of patients with an unplanned contact with hospital within 180 days after inclusion stay discharge

- Number of unplanned contacts with hospital per patient within 12 months after inclusion stay discharge

- Time to next unplanned contact with a hospital within 12 months after inclusion stay discharge

Secondary inclusion stay outcomes were:

- Proportion of patients not hospitalized following admission to the ED (patients with conditions resolved in the ED)

- Length of stay at the ED

- Overall length of hospital stay

- Efficiency of the intervention (working model for conducting medication reconciliation and medication review in the ED-setting)

Amendment to study outcomes after the study commenced, although before any outcome data files were available:

- Proportion of patients with an unplanned contact with hospital:
  - within 90 days after inclusion stay discharge
  - within 30 days after inclusion stay discharge

It was decided to add these amendments due to the relatively short intervention compared with the long follow-up time.

## Sample size calculation

The sample size calculation was based on an expected readmission frequency of 50% during 12 months following inclusion [45], which is also in line with Norwegian readmission frequency. A 10% reduction in hospital readmissions was defined as a clinically relevant effect of the intervention, which was considered realistic according to earlier studies and accounting for our general population [32]. Accordingly, it was calculated that 385 patients would have to be included into each group, with significance level of 5% and study power of 80%. To compensate for dropouts, it was decided to include 400 patients in each project group, i.e., a total of 800 patients.

## Statistical analysis

Data handling were conducted in EpiData manager and EpiData entry client 4.4.3.1 r691, EpiData Software [48]. Statistical analyses were carried out in STATA Statistical Software: Release 16., StataCorp LLC 2019. Demographic statistics are given as median and interquartile ranges (IQRs) for continuous variables and as percentage for categorical variables.

Pearson's chi-square test was applied to determine proportion of patients with unplanned contact with hospital during follow-up time (primary outcome), significance level was set to 0.05. Logistic regression was applied to determine odds ratio with 95% confidence interval (CI). The Kaplan-Meier estimator was used to estimate the survival function regarding time to next unplanned contact with hospital (event). Kaplan-Meier curves and log-rank test were applied to present a visual representation and test difference between groups, respectively. *Cox proportional hazards model* was utilized to determine hazard ratios and 95% CI. Negative binomial regression was utilized to compare mean length of stay and number of unplanned hospital events within 12 months after inclusion stay discharge, in the latter individual patient-time in study was applied as exposure-variable. Mann-Whitney U test was applied to compare median length of stay. For the secondary outcomes comparative analyses were explorative.

Analyses of mortality data revealed that 83% of patient who died within 12 months after inclusion stay discharge had an unplanned contact with hospital before date of death. Sensitivity analyses on the primary outcome and the survival analysis to determine time to next contact with hospital was conducted. In sensitivity analyses death was included as competing risk in the outcome measures (both death during hospital stay and within 12 months from inclusion stay discharge) (Table A and Fig A in S4 Appendix). In addition, sensitivity analysis on inclusion stay outcomes was conducted with patients who died during hospital stay (Table B in S4 Appendix). The sensitivity analyses revealed that results did not change and there was no difference between group in survival curve with death as event (Fig B in S4 Appendix). To clarify interpretation of the outcome measures it was therefore decided to excluded patients who died during hospital stay from both inclusion stay analysis and follow-up analysis. Further, in the primary outcome and survival analysis death within 12 months from inclusion stay

discharge was censored. However, in the negative binomial regression, time of death within 12 months from inclusion stay discharge was accounted for through the exposure variable.

Intention-to-treat analysis was conducted on all patients with follow-up data allocated to each group. Per-protocol analysis was conducted on intervention group patients where the intervention medication reconciliation was completed, excluding patients were 1) no drugs were recorded by physicians in the electronical patient record (based on retrospective assessment of admission-note, medication chart and discharge note) and 2) information about regular drugs was omitted by physicians from the electronical patient record for unknown reason (based on retrospective assessment of admission-note and medication chart). All control group patients with follow-up data were included in per-protocol analysis.

## Results

During the data-collection shifts between 24 April 2017 to 16 May 2018, approximately 1900 patients were admitted to the ED at Diakonhjemmet Hospital and were eligible for inclusion. Of these, 831 (43.7%) patients were assessed for eligibility for inclusion (Fig 1), whereas the remaining patients were not assessed due to ED crowding which exceeded study pharmacists' capacity. Eighteen patients were excluded before randomization as they were not capable of providing written, informed consent or declined to participate. A total of 813 patients were included and randomized. After randomization, 6 patients were excluded, leaving 807 patients eligible for inclusion stay analysis, 405 in the intervention group and 402 in the control group (Fig 1).

Mean age for included patients (n = 807) was 65.4 years (±18.5), median age was 69.2 years (IQR 26.6, range 18.7–99.4). A total of 56.8% of the patients were aged over 65 years and 51.7% were men. Table 1 shows demographics for included patients in their allocated group.

Intention-to-treat primary analysis revealed that 201 patients (51.0%) in the intervention group (n = 394) and 210 patients (53.2%) in the control group (n = 395) had an unplanned contact with hospital (ED-visit or hospital admission) within 12 months after inclusion stay discharge. The intervention had no statistically significant impact on the primary outcome, p = 0.546, OR 0.92, 95%CI 0.69, 1.21. According to per-protocol analysis 130 (48.7%) intervention group patients (n = 267) compared with 210 (53.2%) control group patients (n = 395) had an unplanned contact with hospital (ED-visit or hospital admission) within 12 months after inclusion stay discharge. However, the difference was not statistically significant, p = 0.258, OR 0.84, 95%CI 0.61, 1.14.

The intention-to-treat analysis revealed that median time to next unplanned contact with hospital (ED-visit or hospital admission) (Fig 3A) was 330 days for the intervention group and 308 days for the control group (p = 0.755, HR 0.97, 95%CI 0.80, 1.18). In the per-protocol *analysis*, the median time to next unplanned contact with hospital (ED-visit or hospital admission) (Fig 3B) exceeded follow-up time for the intervention group and was 308 days for the control group (p = 0.378, HR 0.91, 95%CI 0.73, 1.13).

For intervention group patient where a drug list was registered by ED physicians at admission (n = 341) the median number of registered drugs were in line with the median number of drugs registered by study pharmacists in the reconciled drug lists (Table 1), and higher than the number of registered drugs per patient in the control group. However, for 64 (15.8%) patients in the intervention group no drug list were registered by physicians at admission (Table 1). Medication reconciliation conducted by study pharmacists revealed that these 64 intervention group patients in fact used median 2 (IQR 5, 0–15) regular drugs. In addition, it was revealed that for 66 (16.3%) of the intervention group patients the drug list was not correct, information regarding one or more of their regular drugs was incorrectly registered by physicians for unknown reason.

**Table 1. Demographics of included patients.**

| Variable | Categories | Intervention group n = 405 | Control group n = 402 |
|---|---|---|---|
| **Age** | *Median, years (IQR, range)* | 67.2 (27.3, 18.7–96.4) | 70.2 (25.1, 19.1–99.4) |
| | *Patients ≥65 years number (%)* | 218 (53.8) | 240 (59.7) |
| **Sex (men)** *number (%)* | | 212 (52.4) | 205 (51.0) |
| **Distribution of referral reasons** *number (%)* | *Medical* | 281 (69.4) | 282 (70.2) |
| | *Surgical* | 123 (30.4) | 119 (29.6) |
| | *Rheumatological[e]* | 1 (0.3) | 1 (0,3) |
| **Triage category** [49] *number (%)* | *Triage 1* | 0 | 1 (0,3) |
| | *Triage 2* | 127 (31.9) | 136 (34.4) |
| | *Triage 3* | 157 (39.5) | 147 (37.2) |
| | *Triage 4* | 111 (27.9) | 110 (27.9) |
| | *Triage 5* | 3 (0,8) | 1 (0,3) |
| **Admitted from** *number (%)* | *General practitioner* | 105 (25.9) | 123 (30.6) |
| | *Nursing home* | 10 (2.5) | 12 (3.0) |
| | *Other hospital* | 40 (9.9) | 38 (9.5) |
| | *Municipal emergency room* | 134 (33.1) | 112 (27.9) |
| | *Emergency medical communication centre* | 47 (11.6) | 43 (10.7) |
| | *Directly to emergency department [f]* | 6 (1.5) | 7 (1.7) |
| | *Various [g]* | 63 (15.6) | 67 (16.7) |
| **Living situation before admission** *number (%)* | *Home without help* | 354 (87.4) | 330 (82.1) |
| | *Home with assistant living (home care and/or multidose packed drugs)* | 40 (9.9) | 55 (13.7) |
| | *Nursing home/ Rehabilitation* | 11 (2.7) | 17 (4.2) |
| **Admissions to DH[a] 12 months before inclusion stay admission** | *Median number of admissions (IQR, range)* | 0 (1, 0–26) | 0 (1, 0–28) |
| | *Number of patients with at least one admission (%)* | 128 (31.6) | 148 (36.8) |
| **Tentative referral reasons** [50] **(communicated by referring healthcare personnel)** [b] *number (%)* | *Symptoms, signs, and abnormal clinical and laboratory findings, not elsewhere classified* | 149 (36.8) | 144 (35.8) |
| | *Diseases of the circulatory system* | 88 (21.7) | 93 (23.1) |
| | *Diseases of the digestive system* | 37 (9.1) | 40 (10.0) |
| | *Injury, poisoning, and certain other consequences of external causes* | 31 (7.7) | 31 (7.7) |
| | *Diseases of the respiratory system* | 25 (6.2) | 32 (8.0) |
| **Discharge diagnoses** [50] **(set by the hospital physician discharging the patient)** [b] *number (%)* | *Diseases of the circulatory system* | 115 (28.4) | 120 (29.9) |
| | *Symptoms, signs, and abnormal clinical and laboratory findings, not elsewhere classified* | 84 (20.7) | 70 (17.4) |
| | *Diseases of the respiratory system* | 65 (16.0) | 68 (16.9) |
| | *Diseases of the digestive system* | 57 (14.1) | 56 (13.9) |
| | *Factors influencing health status and contact with health services* | 48 (11.9) | 56 (13.9) |
| | *Endocrine, nutritional, and metabolic diseases* | 48 (11.9) | 43 (10.7) |
| **Number of regular drugs registered by physicians at admission** *per patient with a registered drug list* | *Median regular drugs (IQR, range)* | 4 (IQR 6, 0–16) [h] | 3 (IQR 6, 0–19) |
| **Patients where no drug list was registered by physicians at admission** *number (%)* | | 64 (15.8) | 61 (15.2) |
| **Number of regular drugs registered by study pharmacists at admission** [c] *per intervention group patient* | *Median regular drugs (IQR, range)* | 4 (IQR 6, 0–19) | - |
| **Patients receiving pharmacist service in hospital wards (not part of the intervention)** *number (%)* [d] | *Medication reconciliation* | 3 (0.7) | 12 (3.0) |
| | *Medication review* | 63 (15.6) | 48 (11.9) |

*(Continued)*

**Table 1.** (Continued)

| Variable | Categories | Intervention group n = 405 | Control group n = 402 |
|---|---|---|---|
| **Mortality** *number (%)* | *Died during hospital stay* | 6 (1.5) | 2 (0.5) |
| | *Died within 12 months from inclusion stay discharge* | 33 (8.4) | 37 (9.4) |

[a] DH: Diakonhjemmet Hospital

[b] The table presents only the most frequent tentative referral reasons and discharge diagnoses, each patient had 1–3 tentative referral reasons and 1–7 discharge diagnoses

[c] After conducting medication reconciliation as part of the investigated intervention

[d] Pharmacist service on hospital wards was part of standard care at Diakonhjemmet Hospital and was not part of the investigated intervention

[e] Patients with rheumatological referral reasons are seldom admitted to the investigated ED, majority of these patients are admitted elective

[f] Patients arrived directly to the emergency department without referral

[g] Various includes unlimited referral (patients with frequent hospitalization/ or patients with unresolved conditions can be offered this solution, and can contact the hospital directly when needed), patients with complications after recent surgical or medical hospital treatment (within 3 months), elective admissions, transfer from other unit at Diakonhjemmet Hospital, e.g., out-patient clinic, psychiatry unit

[h] Reconciled drug list documented by study pharmacist was available when physicians documented drug lists for intervention group patients

A total of 646 drug-related problems were identified and documented by study pharmacists through medication review. The interdisciplinary team assessed 23.1% of the drug-related problems to be clinically relevant to identify during the ED stay. Among these the most frequent drug-related problems were adverse effects (25.5%), drug monitoring (10.1%), adherence issues (9.4%) and drug-interactions (9.4%). Further, 50.9% of the drug-related problems were found to be clinically relevant during the hospital stay, whereas 26.0% were not clinically relevant to the patients' treatment during hospital stay. Physicians implemented the study pharmacists' recommendations for 44.8% of the drug-related problems where the interdisciplinary team concluded clinically relevant to identify during the ED or hospital stay.

Further analysed secondary outcomes is presented in Table 2.

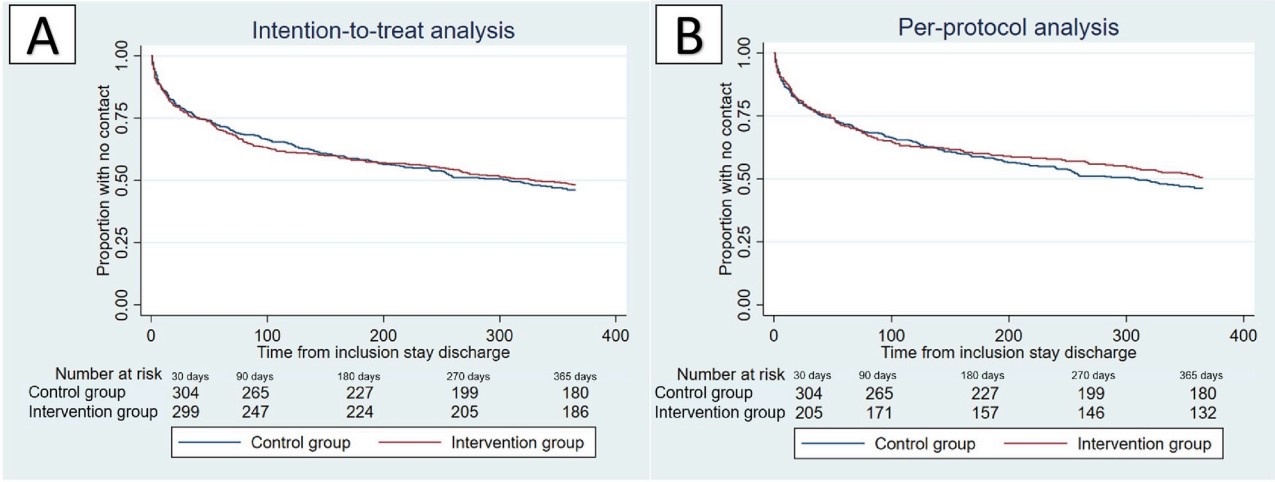

**Fig 3. Time to next unplanned contact with hospital (ED-visit or hospital admission). (A) Intention-to-treat analysis** of patients with follow-up data, intervention group (n = 394) vs. control group (n = 395) **(B) Per-protocol analysis** of patients with follow-up data, intervention group (n = 267) vs. control group (n = 395).

**Table 2. Secondary outcomes (Intention-to-treat analysis) comparing intervention- and control group patients.**

| Inclusion stay endpoints | | Intervention group | Control group | P-value |
|---|---|---|---|---|
| | | *n = 399* | *n = 400* | |
| *Patients not hospitalized* | *Number (%)* | 129 (32.3%) | 130 (32.5%) | 0.959 [a] |
| *Length of emergency department stay,* | *Median, hours (IQR, range)* | 3.1 (2.1, 0.6–10.6) | 3.0 (1.9, 0.6–8.9) | 0.079 [b] |
| | *Mean, hours (±SD)* | 3.5 (±1.5) | 3.3 (±1.4) | 0.119 [c] |
| *Length of hospital stay* | *Median, days (IQR, range)* | 1.0 (2.0, 0.0–37.8) | 1.0 (2.7, 0.0–34.1) | 0.730 [b] |
| | *Mean, days (±SD)* | 2.1 (±4.1) | 1.7 (±2.9) | 0.073 [c] |
| **Follow-up endpoints** | | *n = 394* | *n = 395* | |
| *Patients in contact with hospital within 180 days from inclusion stay discharge* | *Number (%)* | 163 (41.4%) | 163 (41.3%) | 0.976 [a] |
| *Patients in contact with hospital within 90 days from inclusion stay discharge* | *Number (%)* | 141 (35.8%) | 125 (31.7%) | 0.219 [a] |
| *Patients in contact with hospital within 30 days from inclusion stay discharge* | *Number (%)* | 89 (22.6%) | 87 (22.0%) | 0.849 [a] |
| *Number of contacts with hospital within 12 months from inclusion stay discharge* | *Median, IQR, range* | 1 (2, 0–34) | 1 (2, 0–28) | 0.523 [c] |

Inclusion stay endpoints are presented for all patients who survived the inclusion stay. Follow-up endpoints are presented for all patients with available follow-up data.

[a] P-values generated with Pearson's chi-square test

[b] P-values generated with Mann-Whitney U test

[c] P-values generated with negative binomial regression

## Discussion

Pharmacists-led ED medication review did not significantly reduce the proportion of patients with an unplanned contact with hospital compared with standard care in the present study. These results are in line with three recent ED studies investigating similar interventions [51–53]. Santolaya-Perrín et al. [51] conducted a RCT investigating the efficacy of a medication review program in the ED. It was reported that the program had no overall impact on the number of ED revisits and hospital readmissions compared with standard care, no other clinical and post-discharge outcomes was investigated [51]. One arm in a RCT conducted by Graabaek et al. [52] investigated impact of a pharmacist-led medication review in the ED on drug-related readmissions, it was reported that the intervention had no impact on the chosen outcome. A quasi-randomized trial conducted by Hohl et al. [53] investigated clinical impact of pharmacists-led ED medication reviews, and report an 8% reduction in median length of stay in hospital within 30 days from discharge compared with standard care. However, a follow-up study to the quasi-randomized study found that pharmacist-led ED medication review did not result in long-term changes to outpatient health services utilization [54]. The results from our study and the other studies conducted in the ED are however in contrast with RCTs investigating impact of pharmacists-led interventions on clinical outcomes for patients in hospital wards. These in-hospital RCTs have reported reduction in ED revisits and hospital readmissions [32, 33, 35, 39], and further reduction in overall survival [38] as effects of their interventions. The difference in impact between RCTs conducted in hospital wards and studies performed in the ED may be explained by the patient population, the study setting, the degree of acceptance of pharmacists' recommendations and the pragmatic nature of the ED interventions compared with the in-hospital interventions. The RCTs reporting effects on clinical outcomes after in-hospital pharmacist-led intervention investigated specific risk patient groups [32, 33, 35, 38], such as patients older than 80 years [32], patients using specific risk-drugs [33], multimorbid patients [38] or having specific prior diagnoses [35]. The three recent ED studies also included specific patient groups; patients aged over 65 years [51, 52], and patients at high risk of adverse drug events [53]. In the presented RCT we aimed to investigate the impact of the intervention in a general, real-world ED population with both medical and

surgical referral reasons and regardless of patients' age. This led to inclusion of a heterogeneous group of patients. The study population varied widely regarding their need for healthcare utilization post-discharge (0–34 unplanned contacts during 12 months). In addition, patients in need of a hospital admission following their ED stay were included alongside of patient who were directly discharged from the ED, indicating variation in the severity of their acute problems. This diversity in patient population may have obscured the effect of the intervention.

The ED setting is a challenging study setting compared with the hospital ward setting. Average length of stay in the investigated ED was 3.2 hours in 2018, and the limited timeframe may have affected the communication between physicians and study pharmacists necessary for implementation of the intervention. The acceptance of the study pharmacists' recommendations on drug-related problems in our study was similar to the acceptance reported in two of the recent ED studies [51, 52]. However, the acceptance reported by Santolaya-Perrín et al. varied between the investigated study sites (27–53%) [51], and a statistically significant reduction in number of ED revisits and hospital readmissions was reported in the two study sites with the highest acceptance [51]. Hohl et al. [53] and the follow-up study [54] did not investigate this concern, however reported that low acceptance of pharmacists' recommendations could be a study limitation. Earlier RCTs conducted in hospital wards [32, 37, 38] has reported higher acceptance of pharmacists' recommendations compared to our study. Studies identifying non-adherence to pharmacist recommendations as a threat to the success of medication review interventions have been published after data collection of our study was finalized [54, 55]. Despite the low acceptance of study pharmacists' recommendations our explanatory analyses estimated that median time until next unplanned contact with hospital was more than 57 days longer for the patients who received the intervention as planned (per-protocol) compared with standard care. Indicating that the intervention may have prolonged time until next unplanned contact with hospital for those patients where findings from the intervention was used by physicians at admission. These results must be verified by future research.

In the real-world work-chain healthcare professionals have to rely on the next link of the chain for follow-up [56]. The presented study aimed to investigate a pragmatic less resource-demanding intervention than used in prior RCTs conducted in hospital wards [31–34, 38]. Hence, the study pharmacists did no further follow-up after communicating findings from the intervention to the ED physicians responsible for treating the patient and registering the findings in the electronic patient record. The reconciled drug lists documented by study pharmacists were, however, only used by physicians in 66% of the intervention group patients (per-protocol patients). Indicating that the working model for medication reconciliation used in our study, may not represent the most efficient working model in the time-pressured ED setting. A proposal for working model redesign to improve drug information flow is to assign the task of obtaining drug histories to trained clinical pharmacists, and further the pharmacists should document their findings directly in the medication chart [57]. This practice is currently not systematically implemented in Norway.

Regarding drug-related problems identified during ED medication review, it must be considered whether the ED is the suitable setting for identification. In the ED setting, the focus is on the acute problem bringing the patient to hospital, thus preventative long-term drug management decisions may not be prioritized [54]. A recent study conducted at Diakonhjemmet hospital reported that ED physicians conduct numerous essential tasks and distributes a minor proportion of their time on drug-related tasks, on average 7.8 minutes were spent on obtaining and registering a patient's drug list at admission [58]. Pharmacist-led medication reconciliation conducted efficiently in the ED may reduce the physicians' workload [58], however, assessments regarding identified drug-related problems requires dialogue and may be more

challenging in the time-pressured environment of the ED. However, 23.1% of the drug-related problems identified by study pharmacists in our study were found to be clinically relevant to identify during ED visit. In addition, drug-related ED visits are common [12–18, 59], and frequently not recognized by ED physicians in the fast-paced workflow [14, 15, 23]. A sub-study we recently published, investigating the intervention group in the presented RCT, reported that ED pharmacists efficiently flagged drug-related ED visits early during the ED visit [59].

If conducting medication review in the ED, the working model for medication review must be redesigned to fit the ED setting, for instance by focusing mainly on flagging ED visits that could be drug-related and identifying drug-related problems important to solve immediately, such as adverse effects, adherences issues, drug monitoring and interactions. Other identified drug-related problems must be followed-up during hospital stay or by healthcare personnel in the primary healthcare. However, our results indicates that this follow-up cannot solely rely on the next link of the chain. A referral of the patient to follow-up by a clinical pharmacist at a hospital ward can be a feasible solution. And for patients directly discharged from ED, the ED pharmacist can write a follow-up note addressed to relevant health care personnel in the primary healthcare [51, 60].

Regarding length of stay, explorative analyses revealed that the median and average length of stay in the ED were longer (6 minutes and 12 minutes longer, respectively) for the intervention group compared with the control group. Further, there was no difference in median length of hospital stay, however, average length of hospital stay was 10 hours longer for the intervention group compared with standard care. Many factors are known to influence length of stay both in ED and in hospital, e.g., physician and nurse staffing, accessibility of medical information, in-hospital capacity, patients' age, and patients' condition [61, 62]. It can, however, not be ruled out that the intervention prolonged the ED and/or hospital stay for some of the patients. These results must be verified by future research.

## Strengths and limitations

This study was an RCT, and there was not observed any differences between demographics in groups at ED admission. Further, no evidence of random differences in standards of care during the hospital stay was observed between study groups. However, given the single study location, in one specific healthcare system (where patients are referred to the ED by healthcare personnel of the primary healthcare system), the results are not necessarily generalizable to EDs in other countries. Three study pharmacists were involved in conducting the intervention and strict guidelines for the deliverance of the intervention were made, which limits the inter-individual variability.

A strength with our RCT is that the chosen outcomes were objective measures, possible to measure with a high degree of precision. However, data for the outcome measures were harvested from routinely collected data (data from the Norwegian Patient Registry), which according to a recent meta-research study [63] makes it more challenging to reveal treatment benefits compared with traditional RCTs not using routinely collected data. Further, a core outcome set with recommended outcomes for studies investigating impact of medication review-interventions was recently published [64]. The core outcome set lists drug-related admissions, outcomes related to drug use (e.g., overuse, underuse) and patient-reported outcomes as recommended [64]. These outcomes are more subjective compared to our chosen outcomes; however, it can be argued that these outcomes may have been more sensitive to the impact of the investigated intervention.

Blinding to group allocation was not possible due to the nature of the intervention. Hence, a spillover effect of the intervention to control group patients cannot be ruled out. In addition,

registration of clinical pharmacists' ward activities revealed that 14.9% of the control groups patients and 16.3% of the intervention group patients received pharmacist service during their hospital stay (at hospital wards), as part of standard care. This may have affected the results and making it more difficult to reveal differences between groups.

Only 41.6% of patients admitted to the ED were assessed regarding eligibility for inclusion. Study pharmacists had no specific criteria for which patients to include in case of ED crowding. Summary statistics from Diakonhjemmet Hospital for the period from 2017 to 2018 reveal that 57.2% of patients admitted to the ED were aged over 65 years, further 73% of patients were referred with a medical referral reason, 27% with a surgical referral reason. Hence, both the annual age-distribution and distribution regarding referral reason were similar to the study population in the presented study. However, selection bias regarding other variables cannot be rejected.

An interdisciplinary team was utilized to assess the clinical relevance of the identified drug-related problems in this study, verifying the importance of the intervention findings. Further, adherence to the intervention findings was registered. Despite designing the study to be sufficiently powered, the number of intervention group patients that could be included in per-protocol analysis did not reach calculated sample size. Hence, the assumptions underlying the sample size computation were not realized for the intervention group in these analyses, and the results should therefore be interpreted with some caution.

## Conclusion

This RCT revealed that ED pharmacist-led medication review as currently designed, did not significantly impact clinical outcomes or post-discharge outcomes. Nevertheless, exploratory analyses revealed promising results for patients receiving the intervention as intended. Our results indicates that future studies in this field could benefit from pharmacists taking on a more active role to ensure higher acceptance of their clinically relevant recommendations. The presented proposals for redesign of the pharmacist-led intervention are that pharmacists should document their findings from medication reconciliation directly in the medication chart. Further, ED pharmacists should focus on flagging ED visits suspected to be drug-related and revealing acute drug-related problems, such as adverse effects, adherences issues, drug monitoring, and interactions, which can be discussed with ED physicians. Patients in need of further evaluation and management of their drug list, should be referred to a targeted pharmacist service in hospital wards. Results from the presented RCT can be used to tailor pharmacist-led medication review to the ED setting in future studies. Additional studies with a redesigned intervention should be conducted to verify the results from the presented RCT.

## Supporting information

**S1 Appendix. CONSORT checklist.**
(PDF)

**S2 Appendix. Original study protocol, protocol amendments, and the timeline of the study with milestones.**
(PDF)

**S3 Appendix. Detailed description of the drug-related problems categories.**
(PDF)

**S4 Appendix. Sensitivity analyses and survival curves.**
(PDF)

## Acknowledgments

The authors thank Ann-Christin Riddarsporre who contributed as study pharmacist, Therese Tran and Ingrid Kinne Tunestveit who summarized patient information presented to the interdisciplinary team, Merethe Nilsen and Aasmund Godø who contributed as members of the interdisciplinary team, employees at the emergency department and physicians affiliated to the emergency department during the data collection for the positive attitude to the study, and finally, Andrew H. Reiner at Oslo Center for Biostatistics and Epidemiology for valuable support during the preparation of outcome data files.

## Author Contributions

**Conceptualization:** Lisbeth Damlien Nymoen, Tron Anders Moger, Erik Øie, Espen Molden, Kirsten Kilvik Viktil.

**Formal analysis:** Lisbeth Damlien Nymoen, Tron Anders Moger.

**Funding acquisition:** Lisbeth Damlien Nymoen, Kirsten Kilvik Viktil.

**Investigation:** Lisbeth Damlien Nymoen, Trude Eline Flatebø, Erik Øie, Kirsten Kilvik Viktil.

**Methodology:** Lisbeth Damlien Nymoen, Trude Eline Flatebø, Erik Øie, Espen Molden, Kirsten Kilvik Viktil.

**Project administration:** Lisbeth Damlien Nymoen, Kirsten Kilvik Viktil.

**Resources:** Lisbeth Damlien Nymoen, Trude Eline Flatebø.

**Software:** Lisbeth Damlien Nymoen.

**Supervision:** Tron Anders Moger, Erik Øie, Espen Molden, Kirsten Kilvik Viktil.

**Writing – original draft:** Lisbeth Damlien Nymoen, Kirsten Kilvik Viktil.

**Writing – review & editing:** Lisbeth Damlien Nymoen, Trude Eline Flatebø, Tron Anders Moger, Erik Øie, Espen Molden, Kirsten Kilvik Viktil.

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
