## [Decision Letter · Decision Letter 0]

24 Mar 2022

PONE-D-22-04157Impact of systematic medication review in emergency department on patients’ post-discharge outcomes -a randomized controlled clinical trialPLOS ONE

Dear Dr. Lisbeth Damlien Nymoen

Thank you for submitting your manuscript to PLOS ONE. After careful consideration, we feel that it has merit but does not fully meet PLOS ONE’s publication criteria as it currently stands. Therefore, we invite you to submit a revised version of the manuscript that addresses the points raised during the review process.

We look forward to receiving your revised manuscript.

Kind regards,

Larry Allan Weinrauch, MD

Academic Editor

PLOS ONE

Journal Requirements:

Additional Editor Comments:

In determining the best use of our pharmacy colleagues as part of the treatment team what have you discerned from this interaction?

Does pharmacy led parallel intervention trigger meaningful interaction or discussion with the ED team?

This effort was a "gatekeeper in" effort, and meaningful interaction with the team might have been helpful early in the stay, however with handoffs of care within the hospital and outside of the hospital these discussions get lost or fumbled. As such, are our outcomes suggesting an absence of statistical benefit true, or would we be better off with a "gatekeeper out" effort to drive better outcomes? As you point out, practitioner time is limited. How can we best husband our resources?

Reviewers' comments:

Reviewer's Responses to Questions

**Comments to the Author**

1. Is the manuscript technically sound, and do the data support the conclusions?

Reviewer #1: Yes

Reviewer #2: Yes

Reviewer #3: Yes

2. Has the statistical analysis been performed appropriately and rigorously? 

Reviewer #1: Yes

Reviewer #2: Yes

Reviewer #3: Yes

3. Have the authors made all data underlying the findings in their manuscript fully available?

Reviewer #1: Yes

Reviewer #2: Yes

Reviewer #3: Yes

4. Is the manuscript presented in an intelligible fashion and written in standard English?

Reviewer #1: Yes

Reviewer #2: Yes

Reviewer #3: Yes

5. Review Comments to the Author

Reviewer #1: This is a well-conducted and designed clinical trial with unfortunately negative results. Sample size and randomization are clearly delineated, and statistical plan both per-protocol and intent-to-treat are clearly stated with proper techniques. I have a few minor comments:

1. careful copyediting: Change "Statistics" to "Statistical Analysis" in the section heading; numerous errors (e.g., Pearson chi should be Pearson's chi-square test)

2. EMA guidelines no longer support the use of p-values to determine balance in baseline tables. This is because a p-value>.05 does not mean the null hypothesis is true, along with the issue of multiple testing since .05 is meaningless with so many tests.

3. Again, your statement that significance is set at .05 has no meaning except perhaps for the primary outcome analysis, which is confirmatory, because of the multiple testing issue. Secondary outcomes are either exploratory or there should be a Bonferroni-type correction.

4. In the conclusions, it should be stated as to whether the assumptions underlying the sample size computation were realized (not the sample size itself, but the assumptions).

Reviewer #2: TO THE AUTHORS:

The group presented an original investigation examining pharmacist-led efforts to improve medication safety during emergency department visits in this analysis. This study represents a continuation of the prior work in terms of scholarly work around the scientific concepts around medications safety within the emergency department environment. I would recommend revising the text to include a specific message, including more data from your cohort, and further refining the presented materials. Nevertheless, the aim of the manuscript should be to clarify the observation within the context of the submitted data and the specifics of the population selection and hopefully more work in the future around specific diagnosis or service line to reduce the heterogeneity of the targeted admissions. Furthermore, the conclusion should focus on extrapolating the results to positively refine the current work and positively impact the care. The presented manuscript could carry a different message regarding the importance of pharmacist-led efforts and the lessons learned for future initiatives. In addition, I have the following suggestions.

Suggestions:

• Although the rationale for this study and the primary hypothesis is stated clearly, I would recommend revising the introduction and focusing on the unique role of the pharmacist during a hospital visit, see https://pubmed.ncbi.nlm.nih.gov/?term=pharmacist+led+intervention+medication+reconciliation+admission&sort=pubdate&size=200

• Please provide the number of patients that were considered for the study before they were assessed for eligibility. Please add the number to both the results section and the figure.

• Please state clearly why hospitalization to medical or surgical wards was not chosen as the point of randomization but rather all comers to the emergency department, which could include those with high risk and low risk and a vast array of diagnoses

• Please provide the clinical characteristics of the population in both arms and compare the number of medications or categories across the two arms

• Please state the software name that was used for the analysis

• I would recommend shortening the discussion

• Please provide a statement about any possible variation of clinical management after the intervention, i.e., is there any reason for the reader to think that the patients in the intervention arm had different standards of care randomly than those in the control arm? In other words, did the intervention of pharmacist-led affect future decisions? during the emergency department stay that could lead to downstream effects of differentiation in the provided quality of care.

• Please provide more evidence for choosing your specific endpoints using the presented one; the hospitalization is a specific endpoint; however, it does not capture GP calls, admissions without emergency department visits.

Reviewer #3: This is an interesting paper on a randomised trial of pharmacist chart review in patients admitted to the emergency department. This is a complex sort of trial to run and a little harder than the usual drug trial, so the authors are to be commended on what they have done here. As with this sort of trial generally ,the reporting can get complex and as such I have a few areas of clarification:

1. Choice of endpoint - the idea of an unplanned hospital contact makes a lot of sense from a cost and resource point of view, as well as the appropriate severity. One presumes that causality is so woolly that it becomes hard to get to the most sensitive outcome which would be clinically relevant medication error. However, excluding deaths is probably not necessarily appropriate; death is worse than hospitalisation, and certainly unplanned, so at the very least an analysis showing no difference in interpretation if the deaths are included would be important - excluding these patients isn't according to the principle of ITT and the missing data is informatively missing.

2. The authors are to be commended on putting sufficient information in the sample size calculation to make it reproducible

3. Subversion of randomisation - envelopes are of course supremely subvertable; can the authors please confirm that the envelopes were opened in numbered sequence? What was put in place to stop a person opening the envelope before deciding whether or not to enrol? What was the average block length as it is this and not the variability that affects guessing next allocation (see the numerous papers on this)

4. Please do not test the baseline characteristics - if the randomisation works appropriately any differences MUST be down to chance.

5. The confidence intervals for effect do not rule out a 30% reduction in hospital contact - does this not imply the results are inconclusive here

6. It's unclear to me the extent to which pharmacists identify issues as part of routine as opposed to only during the sort of intensive note review here. This could be made clearer - is there also evidence of complete adherence to study protocol in both arms? Is there any chance of dilution of effect because of either a halo effect or because of time pressures lowering adherence? This may be in the bottom of page 17, but it would help to understand whether these pharmacy interventions largely replicated the intervention group and found nothing different (i.e. adherence was about 85% for control and 100% for intervention?

6. PLOS authors have the option to publish the peer review history of their article (what does this mean?). If published, this will include your full peer review and any attached files.

Reviewer #1: No

Reviewer #2: No

Reviewer #3: No

---

## [Author Response · Author response to Decision Letter 0]

1 Jul 2022

Thank you for the invitation to submit a revised version of our manuscript to PLOS ONE. 

In this letter we respond to each point raised by the academic editor and reviewers during the review process. And the manuscript is thoroughly revised to clarify and elaborate the addressed topics. The page references provided in this letter refers to the revised manuscript without tracked changes (“Manuscript”).

We sincerely thank the Academic Editor and all three reviewers for providing us with constructive and insightful comments to clarify the interpretation of our findings and improve our manuscript to meet PLOS ONE’s publication criteria.

During the manuscript revision we have ensured that our manuscript meets PLOS ONE's style requirements. In addition, in the revised cover letter we have explained the details regarding ethical and legal restrictions constraining our sharing of the data set utilized for analysis in this manuscript. Changes in reference list can be found after response to reviewers.

Thank you for considering our revised manuscript.

Additional Editor Comments:

In determining the best use of our pharmacy colleagues as part of the treatment team what have you discerned from this interaction?

- Our study revealed that pharmacists-led interventions have to be tailored to the fast-paced workflow of the emergency department. We revealed low acceptance of pharmacists’ recommendations in our study compared with similar in-hospital interventions, however, in line with recent studies investigating pharmacists-led interventions in the emergency department (Santolaya-Perrín et.al. 2019, Graabæk et.al. 2018). Low acceptance of pharmacists’ recommendations has recently been identifying as a threat to the success of medication review interventions (Lisby et.al. 2018, Kitchen et.al. 2020). 

- Based on the experience from the presented study, our proposals for redesign of emergency department pharmacist-led interventions are presented in the revised manuscript (in the discussion section, se specific pages below, and conclusion, page 25). For the treatment team and the patient to take full advantage of the available pharmacists’ resources, pharmacist must take on an even more active role in the emergency department and actively refer patients to follow-up: 

 - Pharmacists should document their findings from medication reconciliation directly in the medication chart (page 22). 

 - Further, sort out and prioritize acute drug-related problems which can be discussed with emergency department physicians, and make a referral to a targeted pharmacist service in hospital wards for handling other drug-related problems (page 22-23).

 - Flag emergency department visits that could be drug-related early during the emergency department stay (Nymoen et.al. 2022), which can be discussed with the emergency department physicians or referred to follow-up by an in-hospital pharmacist. 

Does pharmacy led parallel intervention trigger meaningful interaction or discussion with the ED team?

- Based on experience from our RCT and our results we believe that pharmacist-led interventions can trigger meaningful interactions with the ED team. However, it is important that the interventions used by ED pharmacists are tailored to fit the fast-paced workflow in the ED, as described above and highlighted in our revised manuscript. 

This effort was a "gatekeeper in" effort, and meaningful interaction with the team might have been helpful early in the stay, however with handoffs of care within the hospital and outside of the hospital these discussions get lost or fumbled. As such, are our outcomes suggesting an absence of statistical benefit true, or would we be better off with a "gatekeeper out" effort to drive better outcomes? As you point out, practitioner time is limited. How can we best husband our resources?

- Despite our negative results regarding clinical impact in the presented study, we believe that a pharmacist-led “gatekeeper in” effort has several advantages compared with a “gatekeeper out” effort, especially if combined with targeted pharmacist service in hospital wards as suggested in the revised manuscript (page 22-23). The introduction and discussion section in the manuscript is revised thoroughly to elaborate the advantages of an “gatekeeper-in” effort. 

- We agree that a “gatekeeper out” effort may drive better outcomes, however we believe that with a redesigned “gatekeeper in”-intervention as proposed in our revised manuscript (page 22-23), both increased patient safety during a hospital stay and impact on post-discharge outcomes may be achievable. 

Reviewers' comments:

- We are pleased that none of the reviewers had objections or comments regarding the listed questions.

Reviewers' comments:

Reviewer's Responses to Questions

Comments to the Author

1. Is the manuscript technically sound, and do the data support the conclusions?

- Reviewer #1: Yes

- Reviewer #2: Yes

- Reviewer #3: Yes

2. Has the statistical analysis been performed appropriately and rigorously?

- Reviewer #1: Yes

- Reviewer #2: Yes

- Reviewer #3: Yes

3. Have the authors made all data underlying the findings in their manuscript fully available?

- Reviewer #1: Yes

- Reviewer #2: Yes

- Reviewer #3: Yes

4. Is the manuscript presented in an intelligible fashion and written in standard English?

- Reviewer #1: Yes

- Reviewer #2: Yes

- Reviewer #3: Yes

5. Review Comments to the Author

Reviewer #1

This is a well-conducted and designed clinical trial with unfortunately negative results. Sample size and randomization are clearly delineated, and statistical plan both per-protocol and intent-to-treat are clearly stated with proper techniques. 

- Thank you, Reviewer #1, for acknowledging our effort to clearly delineate the methodology and statistical plan of the presented study

I have a few minor comments:

1. careful copyediting: Change "Statistics" to "Statistical Analysis" in the section heading; numerous errors (e.g., Pearson chi should be Pearson's chi-square test)

- This is corrected in the revised manuscript (page 13).

2. EMA guidelines no longer support the use of p-values to determine balance in baseline tables. This is because a p-value>.05 does not mean the null hypothesis is true, along with the issue of multiple testing since .05 is meaningless with so many tests.

- We agree with this statement and the tests of baseline characteristics is removed from the revised manuscript (table 1, page 16, and throughout the manuscript). 

3. Again, your statement that significance is set at .05 has no meaning except perhaps for the primary outcome analysis, which is confirmatory, because of the multiple testing issue. Secondary outcomes are either exploratory or there should be a Bonferroni-type correction.

- We agree, this is rephrased in the “Statistical Analysis” section (page 13) and revised throughout the results section. 

4. In the conclusions, it should be stated as to whether the assumptions underlying the sample size computation were realized (not the sample size itself, but the assumptions).

- In the revised manuscript this is rephrased to align with this comment (page 25).

 

Reviewer #2

TO THE AUTHORS:

The group presented an original investigation examining pharmacist-led efforts to improve medication safety during emergency department visits in this analysis. This study represents a continuation of the prior work in terms of scholarly work around the scientific concepts around medications safety within the emergency department environment. I would recommend revising the text to include a specific message, including more data from your cohort, and further refining the presented materials. Nevertheless, the aim of the manuscript should be to clarify the observation within the context of the submitted data and the specifics of the population selection and hopefully more work in the future around specific diagnosis or service line to reduce the heterogeneity of the targeted admissions. Furthermore, the conclusion should focus on extrapolating the results to positively refine the current work and positively impact the care. The presented manuscript could carry a different message regarding the importance of pharmacist-led efforts and the lessons learned for future initiatives.

- Thank you, Reviewer #2, for recognizing that our study is a continuation of prior work within the topic of drug safety in emergency department. 

- The manuscript has been thoroughly revised to clarify our results, experiences, and lessons learned. Further, the conclusion has been revised to underline the knowledge gained through the presented RCT, which is valuable when designing future studies in this field (page 25).

- Our RCT was designed and powered to investigate a general emergency department population, thus our data were analyzed as originally intended. Sub-group analyses of specific patient-groups could have been interesting. However, as the study was not designed and powered for this, the results from such sub-group analyses may have been unreliable and affected by many factors (Brookes ST, Whitley E, Peters TJ, Mulheran PA, Egger M, Davey Smith G. Subgroup analyses in randomised controlled trials: quantifying the risks of false-positives and false-negatives. Health Technol Assess. 2001;5(33):1-56. doi: 10.3310/hta5330). No changes are conducted regarding this issue in the revised manuscript.

In addition, I have the following suggestions.

Suggestions:

Although the rationale for this study and the primary hypothesis is stated clearly, I would recommend revising the introduction and focusing on the unique role of the pharmacist during a hospital visit, see https://pubmed.ncbi.nlm.nih.gov/?term=pharmacist+led+intervention+medication+reconciliation+admission&sort=pubdate&size=200

- We agree, in the revised manuscript the introduction section is elaborated to highlight the unique roles of the pharmacist during a hospital stay, and also within the emergency department setting (page 3-4). 

Please provide the number of patients that were considered for the study before they were assessed for eligibility. Please add the number to both the results section and the figure.

- As stated in the original manuscript, all patients admitted to the emergency department during the data-collection shifts were in principle eligible for inclusion (1900 patients). Unfortunately, due to occasionally ED crowding which exceeded study pharmacists’ capacity only 831 of these were assessed for eligibility with regards to willingness and capability of providing written, informed consent. In the revised manuscript we have rephrased to clarify (page 14) and added the total number of eligible patients to Fig. 1 Flow chart of patients eligible for inclusion in the Emergency Department (ED) (page 5).

Please state clearly why hospitalization to medical or surgical wards was not chosen as the point of randomization but rather all comers to the emergency department, which could include those with high risk and low risk and a vast array of diagnoses

- When designing our study, no prior studies had investigated the impact on clinical outcomes of pharmacist-led medication review interventions in emergency department setting. Thus, prior literature did not provide certain evidence with regards to what patient groups could benefit from such interventions. Introducing specific inclusion criteria (for instance medical or surgical referral reason, or only patients over 65 years) could have decreased the heterogeneity of the included patients, however possible valuable information of some patient groups would have been lost. A justification for this decision is added in the introduction of the revised manuscript (page 4).

Please provide the clinical characteristics of the population in both arms and compare the number of medications or categories across the two arms

- In the original manuscript comparison of clinical characteristics such as number of medications and mortality during study period were presented for both arms in text. However, we agree that it is clarifying if these data are presented in table form. In the revised manuscript clinical characteristics for the population is presented for both arms in addition to demographics in Table 1 (page 16-17).

Please state the software name that was used for the analysis

- We used Stata Statistical Software: Release 16 for the analyses, which was stated in the Statistical Analysis section of the original manuscript. To clarify we have elaborated the Software information in the revised manuscript (page 13). 

I would recommend shortening the discussion

- Through the revision of the manuscript, we have aimed to present and discuss our findings as concisely as possible. 

Please provide a statement about any possible variation of clinical management after the intervention, i.e., is there any reason for the reader to think that the patients in the intervention arm had different standards of care randomly than those in the control arm? In other words, did the intervention of pharmacist-led affect future decisions? during the emergency department stay that could lead to downstream effects of differentiation in the provided quality of care.

- This issue was our main hypothesis for this study (presented in the original protocol: Appendix S2). We hypothesized that if we obtained an accurate medication history (through medication reconciliation) and addressed clinically relevant drug-related problems (through medication review) early during the emergency department stay, this would have positive downstream effects. For instance, lead to a more efficient emergency department- and hospital stay, due to treatment decisions being made based on the correct drug list, and in addition identified drug-related problems could be solved before the patient was discharged. And as a sum of these positive downstream effects, we assumed to reveal an impact on clinical outcomes such as unplanned contact with hospital and time to next unplanned contact. 

 - A statement regarding that we observed no evidence of random difference in standards of care during the hospital stay between the intervention arm and the control arm is added in the revised manuscript (page 23). 

- In retrospect we can see that we relied too much on the next link in the treatment chain to follow up our intervention findings, and that is why we have recommended redesigning the intervention for future research in our manuscript (page 22-23 and page 25). 

Please provide more evidence for choosing your specific endpoints using the presented one; the hospitalization is a specific endpoint; however, it does not capture GP calls, admissions without emergency department visits.

- A justification for chosen outcomes is added in the method section of the revised manuscript (page 11). In addition, we have added a discussion of strengths and limitations of our chosen endpoints (page 24). 

- In the revised manuscript we have elaborated what data we harvested from the Norwegian Patient Registry (all hospitalizations, including both isolated emergency department visits and hospital admissions with or without emergency department visits) (page 6). Data on GP calls and GP visits are not included in the Norwegian Patient Registry. However, contact with GP can be challenging to interpretate as an outcome, as this can both indicate worsening of patients’ conditions and appropriate follow-up of patients, whereas unplanned contact with hospital (which was chosen as outcome) clearly indicates an acute worsening in patients’ condition. 

- The revised S2 Appendix comprise an explanation of the discrepancy between what was stated in the clinicaltrials.gov filling and the presented primary outcome.

 

Reviewer #3 

This is an interesting paper on a randomised trial of pharmacist chart review in patients admitted to the emergency department. This is a complex sort of trial to run and a little harder than the usual drug trial, so the authors are to be commended on what they have done here. 

- Thank you, Reviewer #3, for recognizing the complexity of the study and for acknowledging our effort. 

As with this sort of trial generally, the reporting can get complex and as such I have a few areas of clarification:

1. Choice of endpoint - the idea of an unplanned hospital contact makes a lot of sense from a cost and resource point of view, as well as the appropriate severity. One presumes that causality is so woolly that it becomes hard to get to the most sensitive outcome which would be clinically relevant medication error. However, excluding deaths is probably not necessarily appropriate; death is worse than hospitalisation, and certainly unplanned, so at the very least an analysis showing no difference in interpretation if the deaths are included would be important - excluding these patients isn't according to the principle of ITT and the missing data is informatively missing.

- We agree that death is a competing risk to unplanned contact with hospital. In the revised manuscript we have clarified why death were censored in analyses (page 13-14). 

- In the revised S4 Appendix Sensitivity analyses we have added sensitivity analyses on the primary outcome with overall death as a competing risk to unplanned contact with hospital. In line with the previous submitted sensitity analyses (Original S4 Appendix), the additional sensitivity analyses revealed that our results did not change. 

2. The authors are to be commended on putting sufficient information in the sample size calculation to make it reproducible

- Thank you for acknowledging our effort to make this section unambiguous.

3. Subversion of randomisation - envelopes are of course supremely subvertable; can the authors please confirm that the envelopes were opened in numbered sequence? 

- Yes, the randomization envelopes were assigned to each individual participant and opened in numbered sequence, this is clarified in the revised manuscript (page 8).

What was put in place to stop a person opening the envelope before deciding whether or not to enrol? 

- An operation procedure was developed for the inclusion and randomization procedure in this study. This operation procedure stated that an envelope should be assigned to a participant and opened after inclusion, i.e., after the patient had given written informed consent. All three study pharmacists were acquainted with the inclusion and randomization procedure before study start. No further actions were put in place to prevent the addressed issue. Information regarding the developed procedure is added in the revised manuscript (page 6 and page 8).

What was the average block length as it is this and not the variability that affects guessing next allocation (see the numerous papers on this)

- The average block length was 5.0 ± 2.2 (range 2-8), which is also added in the revised manuscript (page 8).

4. Please do not test the baseline characteristics - if the randomisation works appropriately any differences MUST be down to chance.

- We agree with this statement and the test of baseline characteristics is removed from the revised manuscript (table 1, page 16). 

5. The confidence intervals for effect do not rule out a 30% reduction in hospital contact - does this not imply the results are inconclusive here

- In principle we agree with this comment, however the sample size of this study was calculated based on a superiority study design, and not a non-inferiority study design. Therefore, the presented result only indicates no significant difference, hence failure to show superiority of the intervention. No changes are conducted regarding this issue in the revised manuscript.

6. It's unclear to me the extent to which pharmacists identify issues as part of routine as opposed to only during the sort of intensive note review here. This could be made clearer 

- In the revised manuscript we have added information regarding the study setting to claify routinely pharmacists service vs. RCT intervention in the investigated ED, and further clarified pharmacist service at hospital wards at Diakonhjemmet Hospital (part of standard care) (page 6-7).

Is there also evidence of complete adherence to study protocol in both arms? Is there any chance of dilution of effect because of either a halo effect or because of time pressures lowering adherence? This may be in the bottom of page 17, but it would help to understand whether these pharmacy interventions largely replicated the intervention group and found nothing different (i.e. adherence was about 85% for control and 100% for intervention?

- Study protocol only covered pharmacist intervention vs. standard care during the ED visit, and adherence to study protocol in both arms is described in Fig.1 Flow chart of patients eligible for inclusion in the Emergency Department (ED) (number of patients receiving allocated intervention and standard care). 

- Pharmacists service at hospital ward was part of standard care during the hospital stay and were not part of the investigated intervention. Information regarding these acitivities was presented because they could have diluted the effect of the intervention as described in limitations (page 24). A relatively low share of the included patients received pharmacist service during their hospital stay (Table 1, page 16). This should not be interpreted as the remaining included patients did not need a pharmacist intervetion, it is rather an result of limited pharmacist resouces in hospital wards at Diakonhjemmet Hospital (page 7). The number of patients receiving medication reconciliation by pharmacists at hospital wards were lower in the intervention group compared with the control group, which may indicate that the intervention medication reconciliation was satisfying. With regards to medication review, a higher number of intervention group patients received this from pharmacists at hospital wards compared with control group patients. This may be coincidental or hospital wards pharmacist could be following up on drug-related problems revealed during the intervention medication review conducted in the ED. In general, medication review can be repeted several times during a hospital stay with new findings every time due to frequent changes in drug treatment during a hospital stay, and rapid changes in the patient’s condition during a hospital stay. 

Changes to reference list:

- Added an reference for the EpiData Software used for data management in the presented study (reference number 48). 

o 48. Christiansen TB and Lauritsen JM. (Ed.) EpiData - Comprehensive Data Management and Basic Statistical Analysis System. Odense Denmark, EpiData Association, 2010-. Http://www.epidata.dk.

- After submitting the original manuscript it was discovered that Lisby et al. 2010 was not conducted in the emergency department but in an acute hospital ward of internal medicine. Hence, before initiating our study no prior RCT had investigated the clinical impact of pharmacists-led interventions in the ED-setting, this is clarified in the revised manuscript (page 4), and the reference to Lisby et al. 2010 (reference number 38) is referenced together with the other prior in-hospital RCTs in the introduction and discussion sections.

o Lisby M, Thomsen A, Nielsen LP, Lyhne NM, Breum-Leer C, Fredberg U, et al. The effect of systematic medication review in elderly patients admitted to an acute ward of internal medicine. Basic Clin Pharmacol Toxicol. 2010;106(5):422-7.

- During the revision of the introduction and disussion several new references were added. In the introduction references were added to highlight the unique roles of the pharmacist during a hospital stay as recommended by Reviewer #2. And to clarify why a gatekeeper-in effort was chosen in the presented RCT (in reference to Editors comment). In the discussion references were added to clarify discussion of our findings and discuss chosen endpoints, as requested by Reviewer #2 and Reviewer #3.

o New references compared with the original manuscript:

o 1. World Health Organization, Transitions of Care: Technical Series on Safer Primary Care. Geneva; 2016. Licence: CC BY-NC-SA 3.0 IGO. Available at: https://apps.who.int/iris/bitstream/handle/10665/252272/9789241511599-eng.pdf [cited 2022 May 16].

o 2. World health Organization, Medication Safety in Transitions of Care. Geneva;2019 (WHO/UHC/SDS/2019.9). Licence: CC BY-NC-SA 3.0 IGO. Available at: https://www.who.int/publications/i/item/WHO-UHC-SDS-2019.9 [cited 2022 April 17].

o 7. Institute for Healthcare Improvements, Medication Reconciliation to Prevent Adverse Drug Events (2018) Availablefrom: http://www.ihi.org/Topics/ADEsMedicationReconciliation/Pages/default.aspx [cited 2022 May 10].

o 12. McLachlan CY, Yi M, Ling A, Jardine DL. Adverse drug events are a major cause of acute medical admission. Intern Med J. 2014;44(7):633-8.

o 13. Zed PJ, Abu-Laban RB, Balen RM, Loewen PS, Hohl CM, Brubacher JR, et al. Incidence, severity and preventability of medication-related visits to the emergency department: a prospective study. CMAJ. 2008;178(12):1563-9.

o 16. Jatau AI, Aung MM, Kamauzaman TH, Rahman AF. Prevalence of Drug-Related Emergency Department Visits at a Teaching Hospital in Malaysia. Drugs Real World Outcomes. 2015;2(4):387-95.

o 17. Al-Arifi M, Abu-Hashem H, Al-Meziny M, Said R, Aljadhey H. Emergency department visits and admissions due to drug related problems at Riyadh military hospital (RMH), Saudi Arabia. Saudi Pharm J. 2014;22(1):17-25.

o 18. Nickel CH, Ruedinger JM, Messmer AS, Maile S, Peng A, Bodmer M, et al. Drug-related emergency department visits by elderly patients presenting with non-specific complaints. Scand J Trauma Resusc Emerg Med. 2013;21:15.

o 25. Mogensen CB, Thisted AR, Olsen I. Medication problems are frequent and often serious in a Danish emergency department and may be discovered by clinical pharmacists. Dan Med J. 2012;59(11):A4532.

o 28. Patanwala AE, Sanders AB, Thomas MC, Acquisto NM, Weant KA, Baker SN, et al. A prospective, multicenter study of pharmacist activities resulting in medication error interception in the emergency department. Ann Emerg Med. 2012;59(5):369-73.

o 29. Rothschild JM, Churchill W, Erickson A, Munz K, Schuur JD, Salzberg CA, et al. Medication errors recovered by emergency department pharmacists. Ann Emerg Med. 2010;55(6):513-21.

o 30. Stasiak P, Afilalo M, Castelino T, Xue X, Colacone A, Soucy N, et al. Detection and correction of prescription errors by an emergency department pharmacy service. CJEM. 2014;16(3):193-206.

o 31. Marconi GP, Claudius I. Impact of an emergency department pharmacy on medication omission and delay. Pediatr Emerg Care. 2012;28(1):30-3.

o 41. Crook M, Ajdukovic M, Angley C, Soulsby N, Doecke C, Stupans I, et al. Eliciting comprehensive medication histories in the emergency department: the role of the pharmacist. Pharm Pract (Granada). 2007;5(2):78-84.

o 42. Unroe KT, Pfeiffenberger T, Riegelhaupt S, Jastrzembski J, Lokhnygina Y, Colón-Emeric C. Inpatient medication reconciliation at admission and discharge: A retrospective cohort study of age and other risk factors for medication discrepancies. Am J Geriatr Pharmacother. 2010;8(2):115-26.

o 43. Kongkaew C, Hann M, Mandal J, Williams SD, Metcalfe D, Noyce PR, et al. Risk factors for hospital admissions associated with adverse drug events. Pharmacotherapy. 2013;33(8):827-37.

o 47. Asmar R, Hosseini H. Endpoints in clinical trials: does evidence only originate from 'hard' or mortality endpoints? J Hypertens Suppl. 2009;27(2):S45-50.

o 52. Graabaek T, Hedegaard U, Christensen MB, Clemmensen MH, Knudsen T, Aagaard L. Effect of a medicines management model on medication-related readmissions in older patients admitted to a medical acute admission unit-A randomized controlled trial. J Eval Clin Pract. 2019;25(1):88-96.

o 59. Nymoen LD, Björk M, Flatebø TE, Nilsen M, Godø A, Øie E, et al. Drug-related emergency department visits: prevalence and risk factors. Intern Emerg Med. 2022.

o 61. Asplin BR, Magid DJ, Rhodes KV, Solberg LI, Lurie N, Camargo CA. A conceptual model of emergency department crowding. Ann Emerg Med. 2003;42(2):173-80.

o 62. Khosravizadeh O, Vatankhah S, Bastani P, Kalhor R, Alirezaei S, Doosty F. Factors affecting length of stay in teaching hospitals of a middle-income country. Electron Physician. 2016;8(10):3042-7.

o 63. Mc Cord KA, Ewald H, Agarwal A, Glinz D, Aghlmandi S, Ioannidis JPA, et al. Treatment effects in randomised trials using routinely collected data for outcome assessment versus traditional trials: meta-research study. BMJ. 2021;372:n450.

o 64. Beuscart JB, Knol W, Cullinan S, Schneider C, Dalleur O, Boland B, et al. International core outcome set for clinical trials of medication review in multi-morbid older patients with polypharmacy. BMC Med. 2018;16(1):21.

- During revision of the introduction, the following references were removed from the revised manuscript:

o Ashcroft DM, Lewis PJ, Tully MP, Farragher TM, Taylor D, Wass V, et al. Prevalence, Nature, Severity and Risk Factors for Prescribing Errors in Hospital Inpatients: Prospective Study in 20 UK Hospitals. Drug Saf. 2015;38(9):833-43

o Blenkinsopp A, Bond C, Raynor DK. Medication reviews. Br J Clin Pharmacol. 2012;74(4):573-80.

---

## [Decision Letter · Decision Letter 1]

7 Sep 2022

Impact of systematic medication review in emergency department on patients’ post-discharge outcomes -a randomized controlled clinical trial

PONE-D-22-04157R1

Dear Dr. Lisbeth Damlien Nymoen,

We’re pleased to inform you that your manuscript has been judged scientifically suitable for publication and will be formally accepted for publication once it meets all outstanding technical requirements.

Kind regards,

Raphael Cinotti, MD, PhD

Academic Editor

PLOS ONE

Reviewers' comments:

Reviewer's Responses to Questions

**Comments to the Author**

1. If the authors have adequately addressed your comments raised in a previous round of review and you feel that this manuscript is now acceptable for publication, you may indicate that here to bypass the “Comments to the Author” section, enter your conflict of interest statement in the “Confidential to Editor” section, and submit your "Accept" recommendation.

Reviewer #1: All comments have been addressed

Reviewer #2: All comments have been addressed

Reviewer #3: All comments have been addressed

2. Is the manuscript technically sound, and do the data support the conclusions?

Reviewer #1: (No Response)

Reviewer #2: Yes

Reviewer #3: (No Response)

3. Has the statistical analysis been performed appropriately and rigorously? 

Reviewer #1: (No Response)

Reviewer #2: Yes

Reviewer #3: (No Response)

4. Have the authors made all data underlying the findings in their manuscript fully available?

Reviewer #1: (No Response)

Reviewer #2: Yes

Reviewer #3: (No Response)

5. Is the manuscript presented in an intelligible fashion and written in standard English?

Reviewer #1: (No Response)

Reviewer #2: Yes

Reviewer #3: (No Response)

6. Review Comments to the Author

Reviewer #1: (No Response)

Reviewer #2: All my comments were addressed. Congratulations on this scholarly work. Please make sure to upload all Appendix materials as stated.

Reviewer #3: (No Response)

7. PLOS authors have the option to publish the peer review history of their article (what does this mean?). If published, this will include your full peer review and any attached files.

Reviewer #1: No

Reviewer #2: No

Reviewer #3: No

---

## [Editor Report · Acceptance letter]

9 Sep 2022

PONE-D-22-04157R1 

Impact of systematic medication review in emergency department on patients’ post-discharge outcomes -a randomized controlled clinical trial 

Dear Dr. Nymoen:

I'm pleased to inform you that your manuscript has been deemed suitable for publication in PLOS ONE. Congratulations! Your manuscript is now with our production department. 

Kind regards, 

on behalf of

Dr. Raphael Cinotti 

Academic Editor

PLOS ONE